# Epidemiology and burden of focal segmental glomerulosclerosis among United States Veterans: An analysis of Veteran's Affairs data

**Deborah Goldschmidt[1], Mark E. Bensink[2]\*, Zheng-Yi Zhou[1], Sherry Shi[1], Yilu Lin[3,4], Lizheng Shi[3,4]**

**1** Analysis Group, Boston, Massachusetts, United States of America, **2** Travere Therapeutics, Inc., San Diego, California, United States of America, **3** Tulane University, New Orleans, Louisiana, United States of America, **4** Southeast Louisiana Veterans Health Care System, New Orleans, Louisiana, United States of America

\* mark.bensink@travere.com

## Abstract

### Introduction

Focal segmental glomerulosclerosis (FSGS) is a rare glomerular disease that can lead to reduced kidney function and kidney failure (KF). The objective of this study was to describe the epidemiology, characteristics, clinical outcomes, healthcare resource utilization, and costs associated with focal segmental glomerulosclerosis (FSGS) in United States (US) veterans.

### Methods

This retrospective cohort study included patients in the National Veterans Affairs Health Care Network with ≥2 FSGS-associated diagnostic codes that were 30–180 days apart (October 1999–February 2021). Annual FSGS incidence and prevalence per 1,000,000 US veterans were calculated. Patient and disease characteristics as of the index date (date of first FSGS diagnosis) and baseline (6-months pre-index) comorbidities were described. Kaplan-Meier analyses were used to assess overall survival and time from index to KF or death, dialysis, and kidney transplant. Post-index medication use, HRU, and direct health-care costs were summarized.

### Results

The study included 2,515 veterans with FSGS who were followed for an average of 8.9 years. The mean age was 57.5 years, most patients were male (94.6%), and the most common comorbidity was hypertension (87.0%). The mean annual incidence and prevalence of FSGS during 2000–2020 were 19.6 and 164.7 per million veterans, respectively. Approximately half (51.5%) died during follow-up (median time: 11.6 years) and 76.9% had kidney failure (4.1 years). Overall, 43.3% underwent dialysis and 5.8% had a kidney transplant. During follow-up, statins and calcium channel blockers were commonly used (81.9% and

**Data Availability Statement:** Data cannot be shared publicly because the Department of

Veterans Affairs prevents public sharing of national VA EHR data. Researchers with VA appointments can request access to the data through the VA intranet at the VA Data Access Portal. For further assistance, please contact the VA Informatics and Computing Infrastructure (VINCI) at VINCI@va.gov or Yilu Lin at yilu.lin@va.gov.

**Funding:** Funding for this study was provided by Travere Therapeutics, Inc. The sponsor was involved in the study design, data analysis, decision to publish, and preparation of the manuscript.

**Competing interests:** Mark Bensink is an employee of Benofit Consulting, which received consulting fees from Travere Therapeutics, Inc. Debbie Goldschmidt, Zheng-Yi Zhou, and Sherry Shi are employees of Analysis Group, Inc., which has received consulting fees from Travere Therapeutics, Inc. for this work. Yilu Lin and Lizheng Shi have nothing to disclosure. This does not alter our adherence to PLOS ONE policies on sharing data and materials.

75.1%). During the first year post-index, 40% had an inpatient admission and 33% visited the emergency room; mean total healthcare cost per patient in the analysis was $36,543.

## Conclusions

Among US veterans, FSGS is associated with considerable clinical and economic burdens. Better treatments for FSGS are needed to slow kidney disease progression, improve patient outcomes, and reduce the burden.

## Introduction

Focal segmental glomerulosclerosis (FSGS) is a glomerular disease characterized by proteinuria, reduced renal function, and progression to kidney failure (KF) [1]. As a rare disease, FSGS represents approximately 20% and 40% of cases of nephrotic syndrome in children and adults, respectively, with an estimated incidence of 1.4–21 cases per million people globally [2]. It is the most common primary glomerular disorder causing KF in the United States (US), with an estimated prevalence of 4% [3]. As FSGS progresses, it transitions from focal and segmental to more widespread glomerulosclerosis, which results in the loss of integrity of the glomerular filtration barrier and reduction in renal function [1]. Patients who are nephrotic are at greater risk and progress to KF after 5–10 years [3].

FSGS reduces quality of life [4–6] and imposes resource use and economic burden on patients [7, 8]. Once patients have progressed to KF, they require dialysis for years and may qualify to receive a kidney transplant, imposing a substantial clinical and resource burden on patients and health systems [9, 10]. KF also has a negative impact on patients' quality of life, including depression and anxiety, fatigue, daily physical activity, and reported satisfaction with health and function [11–14].

There is currently no cure for FSGS, and no treatment specifically approved for FSGS in the US to effectively prevent or delay progression to KF [15]. Current treatment options include glucocorticoids or calcineurin inhibitors for patients with primary FSGS; for patients with secondary FSGS or FSGS of undetermined cause, treatment for the primary medical condition is recommended. In addition, supportive treatment for the management of persistent proteinuria, including the use of renin-angiotensin system inhibitors such as angiotensin-converting enzyme (ACE) inhibitors and angiotensin II receptor blockers blood pressure control, and dietary salt restriction, are recommended [16].

There are several ongoing studies of treatments for FSGS, including trials of existing immunosuppressive agents as well as novel therapies such as sparsentan, a dual receptor angiotensin receptor type 1 and endothelin type A receptor blocker [15]. With the potential addition of new treatment options for patients with FSGS in the US, understanding the current epidemiology, patient characteristics, treatment use, and clinical and economic burdens associated with FSGS is critical. Using US Veteran's Affairs (VA) data, the objective of this study was to assess these outcomes.

## Methods

### Data source

This study utilized retrospective, deidentified data from the National VA Health Care Network from October 1, 1999 through February 28, 2021. The VA serves over 8 million veterans per

year, with over 1,200 sites of care. The VA stores patient data in the VA Corporate Data Warehouse, including medical encounter information for all medical care administered in VA facilities and costs paid by the VA across sites including medical centers, community-based outpatient clinics, community-living centers, Veteran centers, and domiciliaries. Medical care administered outside of VA facilities and costs paid by other payers, such as Medicare or Medicaid, were not captured. This study received institutional board review approval from the Southeast Louisiana Veterans Health Care System Institutional Review Board on July 2, 2021.

## Study design and sample selection

This was a retrospective study of patients identified as having diagnosis codes associated with FSGS. Patients were included in the estimates of FSGS prevalence and incidence if they had ≥2 diagnosis codes associated with FSGS (International Classification of Diseases, 9th edition, Clinical Modification [ICD-9-CM]: 581.1, 582.1; ICD-10-CM: N03.1, N04.1, N05.1, N06.1, N07.1) that were 30 to 180 days apart. For all other analyses, patients were also required to have at least 6 months of continuous eligibility prior to and at least one year of continuous eligibility following the *index date* (date of the first FSGS diagnosis), and no diagnosis code for COVID-19 at any time during the baseline or follow-up periods (ICD-10-CM code U07.1) (S1 Fig). The *baseline period* was defined as the 6 months before the index date. The *follow-up period* was defined as the period from the index date until death, end of continuous eligibility, or end of data availability, whichever came first.

## Study measures

The raw, unadjusted annual incidence and prevalence of FSGS among US veterans was estimated from 2000–2020. Demographics (age, sex, race/ethnicity, geographic region, and employment status) as of the index date, as well as comorbidities, disease characteristics, and treatments during the baseline period, were summarized. Race/ethnicity based on patient self-report was summarized as it was reported in the VA database. The following categories were summarized: Asian, Black, Hispanic, Native American, Native Hawaiian or Pacific Islander, White Non-Hispanic, and other race. Patients with no race information reported were classified as "Unknown race".

Medication use during years 1, 2, and 3 post-index as well as for the overall follow-up period were summarized. Clinical outcomes included overall survival (OS); incidence of cardio/cerebrovascular events, nephrotic syndrome, and death; and times to KF, dialysis, and kidney transplant. KF was defined as a patient having either a KF diagnosis code (ICD-9/10-CM: 585.6x and N18.6x); chronic kidney disease (CKD) stage 5 (i.e., estimated GFR [eGFR] <15 mL/min/1.73 m$^2$ or ICD-9/10-CM diagnosis codes 585.5x or N18.5); kidney transplant; dialysis; or death.

Healthcare resource use (HRU), including inpatient, outpatient, and emergency room visits, and direct medical and prescription drug costs paid by the VA as reported in the database, were described separately for each of the first two years following the index date. In addition, the number of hospital days among patients with at least one inpatient admission was summarized.

## Statistical analyses

**Epidemiology.** Annual prevalence and incidence counts and rates for FSGS in the VA were calculated from 2000–2020 on a per 1,000,000 persons per year basis. Incidence was estimated as the number of patients with ≥2 diagnoses for FSGS 30–180 days apart, with the first diagnosis in a given year, divided by the number of VA-enrolled veterans in the same year

[17]. Prevalence was estimated as the number of patients with follow-up during the given year who had ≥2 diagnoses for FSGS 30–180 days apart, with the first diagnosis in or before the given year, divided by the number of VA-enrolled veterans in that year [17]. Patients who met the diagnosis criteria but died during the calendar year were also included in the numerator.

**Patient characteristics.** Categorical variables were described using frequencies and percentages, while continuous variables were described using means, medians, and standard deviations (SD). For laboratory values during the baseline period, the measurement closest to the index date was summarized. Urine protein/creatinine (UP/C) ratio measurements were used directly from the data when available; if not, they were calculated using urine protein and creatinine measurements from the same day when possible.

**Clinical outcomes.** Kaplan-Meier (KM) analyses were used to assess OS and time to dialysis, KF, kidney transplant, and death, accounting for censoring at the end of follow-up. KM curves were provided, and median time to event and KM rates with 95% confidence intervals (CI) were calculated. Healthcare costs were adjusted to 2021 US dollars and summarized.

## Results

### Incidence and prevalence

The mean incidence of FSGS from 2000–2020 was 19.6 per million VA-enrolled veterans, starting at 41.1 in 2000 but declining over time, with a low of 9.6 in 2020 (Fig 1A). The average

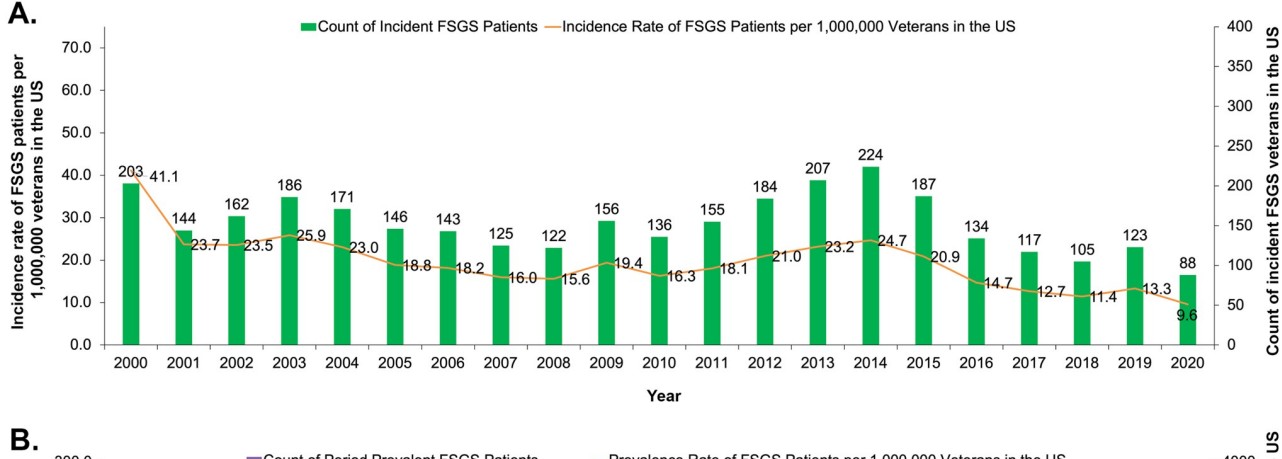

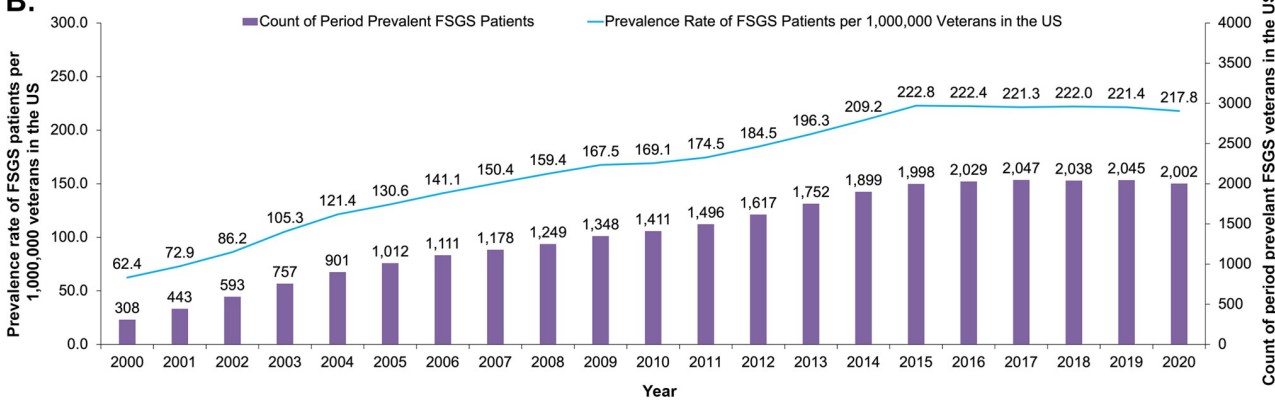

**Fig 1. Estimations of incidence (A) and prevalence (B) of FSGS among US veterans. Abbreviations**: FSGS, focal segmental glomerulosclerosis; US, United States. Data source: Source: US Veteran Affairs Health System Corporate Data Warehouse (October 1999–February 2021).

period prevalence of FSGS was 164.7 per million VA-enrolled veterans per year over the 2000–2020 time period (Fig 1B).

## Patient characteristics

A total of 2,515 patients in the VA database met all criteria for study inclusion (S1 Fig). The demographic, baseline, and clinical characteristics of patients with FSGS-related diagnoses in the VA database are listed in Table 1. The mean age at index was 57.5 years (SD: 13.2; median: 58.7), with 5.4% of patients being female. Most patients were White Non-Hispanic (46.4%) or Black (38.1%). Approximately a third (33.5%) of patients were employed, 32.4% were retired, and 31.5% were unemployed as of the index date. Patients most commonly lived in the Midwest (32.4%), West (30.8%), and the South (27.0%). On average, the follow-up time was 8.9 years (SD: 5.6; median: 7.7).

The mean Charlson Comorbidity Index (CCI; excluding renal disease and KF) score during the baseline period was 1.1 (SD: 1.5) and the most common comorbidities were hypertension (87.0%) and hyperlipidemia (56.0%). The most common classes of medications used during the baseline period were statins (55.1%), ACE inhibitors (53.1%), and calcium channel blockers (47.0%). Among the 1,541 (61.3%) patients with a baseline UP/C measurement, mean UP/C was 2.8 g/g (SD: 4.8; median: 0.4). Among the 1,505 (59.8%) with a baseline eGFR measurement, mean eGFR level was 43.9 (SD: 26.6; median: 38.0) mL/min/1.73 m$^2$. During the baseline period, 12.3% of patients were identified as having KF. CKD stage was unknown for 36.4% of patients, and the most commonly identified stage was stage 4 (16.4%).

## Medication use during the follow-up period

The analyses of medication use in Year 2 and 3 post-index included 2,344 patients with $\geq 2$ years of follow-up and 2,152 with $\geq 3$ years of follow-up, respectively (S1 Table). During the entire follow-up period, 81.9% of patients used statins, with 61–66% of patients using statins within each year analyzed (Year 1, 2, and 3 post-index). Other common medications during the entire follow-up period included calcium channel blockers (75.1%), glucocorticoids (70.1%), and ACE inhibitors (69.9%).

## Clinical events during the follow-up period

The most common cardiovascular or cerebrovascular events identified during the entire follow-up period and in Year 1 was congestive heart failure (49.3% and 18.0%, respectively) (Table 2). Over two-thirds (68.9%) of patients experienced nephrotic syndrome during the entire follow-up period and 51.5% of patients died.

## OS analysis

Among the patients (n = 1,296) who died during the follow-up period, the median time to death of 11.6 years (Fig 2). At 5 and 10 years, the KM survival rates were 79.7% (95% CI: 78.0%, 81.2%) and 55.3% (53.1%, 57.5%), respectively.

## Time to event analyses

**First KF or death.** A total of 1,934 (76.9%) patients experienced KF or died during the follow-up period, with a median time to the first event of 4.1 years (Table 3, Fig 3A). The KM rates at years 1, 5, and 10 were 23.1% (95: CI: 21.5%, 24.8%), 54.8% (52.9%, 56.8%), and 73.7% (71.8%, 75.6%), respectively.

**Table 1. Demographic, clinical, and disease characteristics: United States, 2000–2020.**

| | N = 2,515 |
|---|---|
| **Demographic characteristics at the index date[a]** | |
| Age, years, mean (SD) [median] | 57.5 (13.2) [58.7] |
| Female, n (%) | 135 (5.4%) |
| Race/ethnicity[b], n (%) | |
| Asian Non-Hispanic | 24 (1.0%) |
| Black Non-Hispanic | 957 (38.1%) |
| Hispanic | 122 (4.9%) |
| Native American Non-Hispanic | 12 (0.5%) |
| Native Hawaiian or Pacific Islander Non-Hispanic | 18 (0.7%) |
| Unknown race | 214 (8.5%) |
| White Non-Hispanic | 1,168 (46.4%) |
| Employment status, n (%) | |
| Employed | 842 (33.5%) |
| Retired | 816 (32.4%) |
| Unemployed | 791 (31.5%) |
| Unknown | 66 (2.6%) |
| US geographic region, n (%) | |
| Midwest | 815 (32.4%) |
| West | 774 (30.8%) |
| South | 680 (27.0%) |
| Northeast | 246 (9.8%) |
| Length of follow-up[c] in years, mean (SD) [median] | 8.9 (5.6) [7.7] |
| **Baseline characteristics[a]** | |
| CCI excluding renal disease and KF | |
| Mean (SD) [median] | 1.1 (1.5) [0.0] |
| Comorbidities (top 5), n (%) | |
| Hypertension | 2,188 (87.0%) |
| Hyperlipidemia | 1,408 (56.0%) |
| Anemia | 652 (25.9%) |
| Diabetes without chronic complications | 645 (25.6%) |
| Chronic pulmonary disease | 415 (16.5%) |
| Medication use (top 5), n (%) | |
| Statins | 1,387 (55.1%) |
| ACE inhibitors | 1,336 (53.1%) |
| Calcium channel blockers | 1,183 (47.0%) |
| Diuretics | 676 (26.9%) |
| Glucocorticoids | 587 (23.3%) |
| **Clinical characteristics[a]** | |
| Patients with UP/C measurement[d], n (%) | 1,541 (61.3%) |
| UP/C, g/g, mean (SD) [median] | 2.8 (4.8) [0.4] |
| Patients with eGFR[e], n (%), n (%) | 1,505 (59.8%) |
| eGFR level, mean (SD) [median] mL/min/1.73 m$^2$ | 43.9 (26.6) [38.0] |
| KF[f], n (%) | 310 (12.3%) |
| CKD stage[g], n (%) | |
| Stage 1 | 94 (3.7%) |
| Stage 2 | 242 (9.6%) |
| Stage 3 unspecified | 80 (3.2%) |

(*Continued*)

**Table 1.** (Continued)

|  | N = 2,515 |
|---|---|
| Stage 3a | 232 (9.2%) |
| Stage 3b | 291 (11.6%) |
| Stage 4 | 412 (16.4%) |
| Stage 5/ KF | 249 (9.9%) |
| Unknown | 915 (36.4%) |

**Abbreviations:** ACE, angiotensin-converting enzyme; CCI, Charlson comorbidity index; CKD, chronic kidney disease; eGFR, estimated glomerular filtration rate; FSGS, focal segmental glomerulosclerosis; KF, kidney failure; ICD 9/10-CM, International Classification of Diseases, Ninth/Tenth Revision, Clinical Modification; SD, standard deviation; US, United States; UP/C, urine protein/creatinine.

Notes:

[a] The index date was defined as the date of first FSGS diagnosis between October 1, 1999 and February 28, 2021. The baseline period was defined as the 6-month period before the index date. Clinical characteristics recorded within the National VA Health Care Network were assessed during the baseline period, as close as possible to the index date.

[b] Race/ethnicity were summarized based on the data reported in the VA database. Patients with no race or ethnicity information available were categorized as having "unknown race".

[c] The follow-up period spanned from the index date to the end of enrollment, end of data availability, or date of death, whichever occurred first.

[d] UP/C (in g/g) was reported directly from the data if available; if not, 24-hour protein urine was converted to UP/C if available.

[e] eGFR values were summarized based on what was directly provided in the VA laboratory data; no details on the eGFR calculation methodology were available.

[f] KF during the baseline period was identified using any of the following: ICD-9/10-CM diagnosis code N18.6x; 585.6x; CKD stage 5 diagnosis (ICD-9/10-CM codes: N18.5x; 585.5x); eGFR<15; kidney transplant; or dialysis.

[g] CKD stage was identified by eGFR level (mL/min/1.73 m$^2$) or ICD-9/10-CM diagnosis codes.

## First dialysis

Just under half the cohort (43.3%) were observed in the VA data to undergo dialysis over the course of the follow-up period, with the median time to first dialysis being 11.4 years (Table 3, Fig 3B). The KM rate at 1 year was 12.2% (95 CI: 11.0%, 13.5%), at 5 years was 32.2% (30.3%, 34.2%) and at 10 years was 46.8% (44.5%, 49.1%).

*Kidney transplant.* During the follow-up period, 5.8% of patients were observed in the VA data to receive a kidney transplant (Table 3, Fig 3C). The median was not reached, and the KM rates at years 1, 5, and 10 were 1.0% (95% CI: 0.7%, 1.5%), 3.2% (2.5%, 4.0%), and 7.0% (5.9%, 8.4%), respectively.

## HRU during the follow-up period

Among the 2,515 patients included in the analyses, 39.8% had an inpatient admission in the year following the index date, with an average of 19.1 days spent in the hospital during the year (Table 4). Almost all (99.8%) had an outpatient visit, with an average of 33.6 visits per patient during the year. One third (33.0%) of patients had an emergency room visit.

A total of 2,344 patients had ≥2 years of follow-up post-index and were included in the analyses of HRU during the 2nd year post-index. Among the 26.3% of patients with an inpatient admission during Year 2, an average of 17.7 days was spent in the hospital. Outpatient visits were experienced by almost all patients (98.2%), with an average of 30.9 visits during the 2nd year post-index. During Year 2, 28.3% of patients had an emergency room visit.

**Table 2. Clinical events during the follow-up period[a].**

| | 1st Year Post-Index[a] | Follow-up Period[a] |
|---|---|---|
| | N = 2,515 | N = 2,515 |
| Cardiovascular/cerebrovascular event, n (%) | | |
| Cerebrovascular disease diagnosis | 225 (8.9%) | 680 (27.0%) |
| Myocardial infarction | 142 (5.6%) | 546 (21.7%) |
| Stroke/TIA | 89 (3.5%) | 415 (16.5%) |
| Congestive heart failure | 453 (18.0%) | 1,240 (49.3%) |
| Chronic stable/unstable angina | 43 (1.7%) | 206 (8.2%) |
| Atrial fibrillation | 194 (7.7%) | 625 (24.9%) |
| Hospitalization for heart failure | 239 (9.5%) | 820 (32.6%) |
| Hospitalization for thromboembolic event | 75 (3.0%) | 277 (11.0%) |
| Hospitalization for angina | 27 (1.1%) | 120 (4.8%) |
| Percutaneous coronary intervention | 71 (2.8%) | 264 (10.5%) |
| Coronary artery bypass graft | 106 (4.2%) | 377 (15.0%) |
| Nephrotic syndrome[b], n (%) | 1,372 (54.6%) | 1,732 (68.9%) |
| All cause death | - | 1,296 (51.5%) |

**Abbreviations**: FSGS, focal segmental glomerulosclerosis; ICD-9/10-CM, International Classification of Diseases, Ninth/Tenth Revision, Clinical Modification; TIA, transient ischemic attack.

**Notes**:

[a] The follow-up period spanned from the index date (the date of first diagnosis of FSGS) to the end of enrollment date, end of data availability, or date of death, whichever occurred first.

[b] Nephrotic syndrome was identified either with a diagnosis code (ICD-9-CM: 581.x; ICD-10-CM: N04.x) or via lab values (urine protein/creatinine ≥3 g/g or 24-hour urine protein ≥3.5 g/day and serum albumin levels <3.0 g/dL within 5 days apart).

## Healthcare costs during the follow-up period

Among the 2,515 patients included in the analyses, average total healthcare costs during the 1st year following the index date were $36,543 (SD: $88,858) per patient, comprised of pharmacy costs of $5,402 (SD: $15,499) and medical costs of $31,141 (SD: $80,422) (Table 4). Medical costs were driven by outpatient visit costs of $16,072 (SD: $24,463]) and inpatient admission costs of $13,655 ($72,132); the average emergency room costs were $1,415 ($3,630).

Among patients with FSGS with ≥2 years of follow-up (n = 2,344), the average total healthcare costs per patient during the 2nd year post-index were $29,834 (SD: $58,025), with pharmacy costs of $4,440 ($11,238) and medical costs of $25,394 ($52,046). Of the total medical costs, the average costs per patient were $15,657 (SD: $28,456) in outpatient costs, $8,535 ($36,014) in inpatient costs, and $1,202 ($3,533) in emergency room costs.

## Discussion

In this study, we describe the characteristics and clinical and economic outcomes of US veterans with a diagnosis associated with FSGS within the VA health care system, contributing to the sparse body of research reporting on the epidemiology and outcomes of FSGS. We estimated that the mean incidence and prevalence of FSGS was 19.6 and 164.7 per million veterans, respectively, from 2000 to 2020. In comparison, the estimated global incidence of FSGS reported in the literature is 1.4–21 cases per million, with the wide range due to differences across geographical and racial/ethnic groups [2]. The US incidence of FSGS as estimated by Kitiyakara et al. in 2003 was about 7 per 1 million people [18], while the incidence observed in

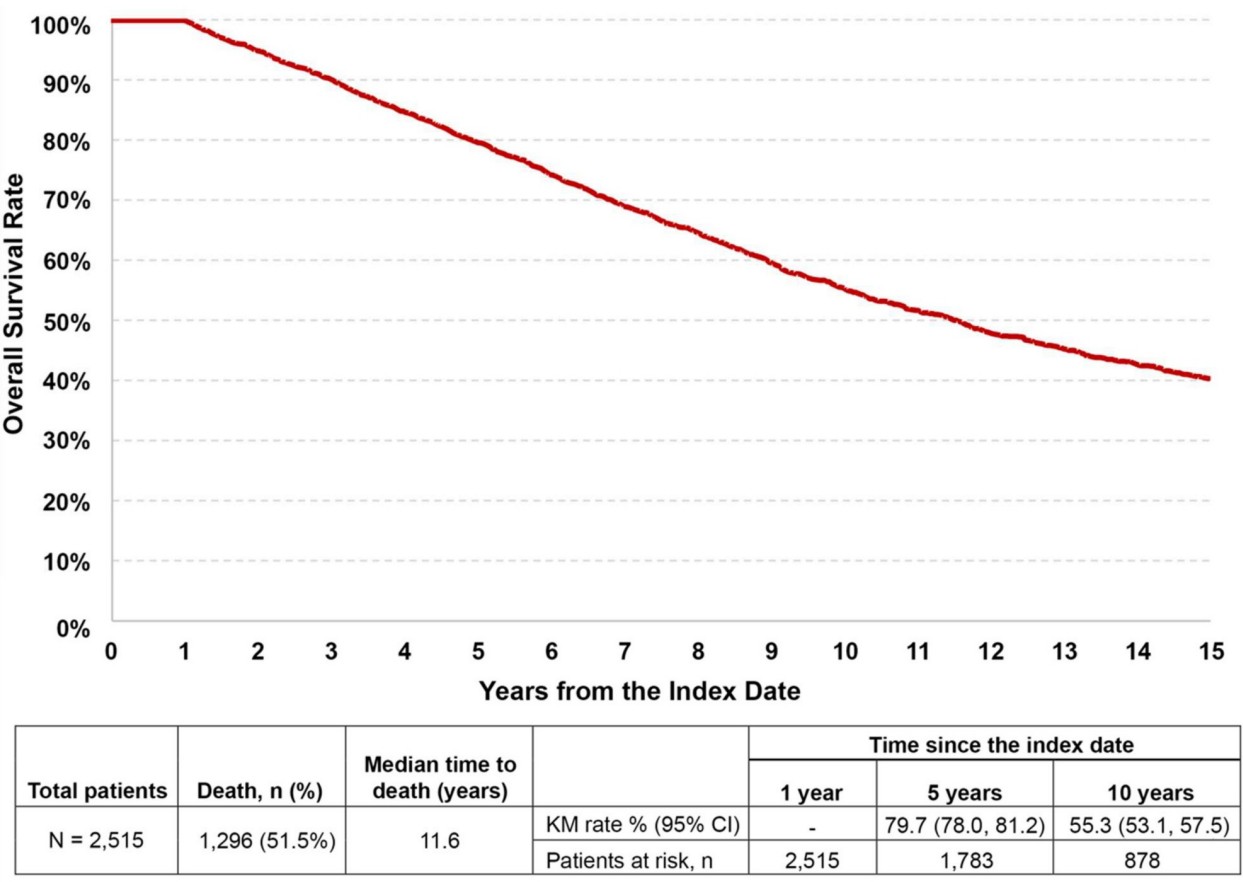

**Fig 2. Overall survival analysis from the index date.** Abbreviations: CI, confidence interval; FSGS, focal segmental glomerulosclerosis; KM, Kaplan-Meier. Note: [a] Survival was measured as the number of days between the index date (date of first diagnosis of FSGS) and the date of death from any cause. Patients who did not experience the event were censored at the end of the follow-up period.

this study was 26 per 1 million US veterans in 2003. The prevalence of FSGS may be higher in the VA population than in the general US population due to the over-representation of patients who are male and of Black/African-American ethnicity, who are known to have higher risk for FSGS [18, 19]. In addition, the reliance on ICD-9/10-CM diagnosis codes and not biopsy for patient identification in this study may result in an overestimation of the incidence and prevalence of FSGS.

FSGS incidence observed in this study varied from 41 per 1 million US veterans in 2000 to 10 per 1 million in 2020. There was less variation in the incidence between 2001–2015; thereafter, incidence generally declined to its lowest point in 2020. The prevalence of FSGS increased over time (from 62 per 1 million veterans in 2000 to approximately 220 per million in 2020), although this was primarily due to new patients being added each year and prior patients remaining in the data. It is possible that the stable/declining incidence and increasing prevalence is related to underestimation of incidence because incident diagnosis may occur outside the VA, which would therefore not be captured in the database. Notably, in 2014, the VA started allowing outside community care for veterans through the Veterans Choice Program due to concerns over long wait times in the VA system. Then, in 2018, the VA MISSION Act replaced the Veterans Choice Program with the Veterans Community Care Program, which further expanded the options for community care. The timing of expansion of community

**Table 3. Time to first KF or death, first dialysis, and first renal transplant during the follow-up period[a].**

|  | Time since index date | | |
|---|---|---|---|
|  | **1 year** | **5 years** | **10 years** |
| Time to first KF or death from the index date[b], KM rate % (95% CI) | 23.1 (21.5, 24.8) | 54.8 (52.9, 56.8) | 73.7 (71.8, 75.6) |
| Time to first dialysis event from the index date[c], KM rate % (95% CI) | 12.2 (11.0, 13.5) | 32.2 (30.3, 34.2) | 46.8 (44.5, 49.1) |
| Time to first renal transplant from the index date[d], KM rate % (95% CI) | 1.0 (0.7, 1.5) | 3.2 (2.5, 4.0) | 7.0 (5.9, 8.4) |

**Abbreviations:** CI, confidence interval; CKD, chronic kidney disease; eGFR, estimated glomerular filtration rate; FSGS, focal segmental glomerulosclerosis; KF, kidney failure; ICD-9/10-CM, International Classification of Diseases, Ninth/Tenth Revision, Clinical Modification; KM, Kaplan-Meier.

Notes:

[a] The follow-up period spanned from the index date (the date of first diagnosis of FSGS) to the end of enrollment date, end of data availability, or date of death, whichever occurred first.

[b] Time to first KF or death was measured as the number of days between the index date (date of first diagnosis of FSGS) and the date of first KF or death during the follow up period. Patients who did not experience the event were censored at the end of the follow-up period. KF or death was defined as a composite measure including KF identified via ICD-9/10-CM diagnosis codes (585.6x and N18.6x), CKD stage 5 (eGFR <15 mL/min/1.73 m$^2$ or diagnosis code [585.5x and N18.5]), kidney transplant, dialysis, and death.

[c] Measured as the number of days between the index date and the date of the first dialysis event during the follow up period. Patients who did not experience the event were censored at the end of the follow-up period.

[d] Measured as the number of days between index date and date of first renal transplant event during follow up period. Patients who did not experience the event were censored at the end of the follow-up period.

care (and opportunities for outside incident diagnosis) for veterans coincides with stable or declining incidence of FSGS and may therefore partially underlie this trend.

Few studies have reported the demographic and clinical characteristics of patients with FSGS so there are limited opportunities for comparison. Among them, the findings have been variable and there are large differences across groups by geographic location, sex, and race/ethnicity [20]. For example, FSGS is 1.5–2 times more common in males vs. females and as much as 5 times more common in Black vs. White patients [2, 18, 21]. However, the main comorbidities observed among the patients in our study are generally consistent with prior literature on commercial or registry populations with FSGS. For example, the most common comorbidity in this study was hypertension (87%), similar to reports by Thomas et al. (74%; a registry of patients in the south-east US) [22], Sim et al. (75%; a commercially-insured population in Southern California) [20], and Tuttle et al. (74%; a cohort study of kidney biopsies from 43 US states) [21]. Additionally, 26% of patients in this study had diabetes, similar to Sim et al. (28%), and the mean eGFR was 43.9 ± 26.6 mL/min/1.73 m$^2$, within the range for patients with FSGS reported by Sim et al. (40.6 ± 28.2 mL/min/1.73 m$^2$) and a claims study using Optum data by Bensink et al. (45.8 ± 39.4 mL/min/1.73 m$^2$) [23]. In all these prior studies, male and Black patients with FSGS were over-represented in comparison with the general US population, although to a lesser extent than in the VA data. The mean age of our patients (58 years) was older than the populations in these studies (i.e., 48 years in Bensink et al., 49 years in Thomas et al. and Tuttle et al., 51 years in Sim et al.), reflecting the generally older age of VA patients.

In this cohort, the median CCI was 1.5, with most patients having hypertension and hyperlipidemia and taking statins or ACE inhibitors during baseline. Furthermore, patients faced a considerable clinical burden, with more than three-quarters developing KF and over half

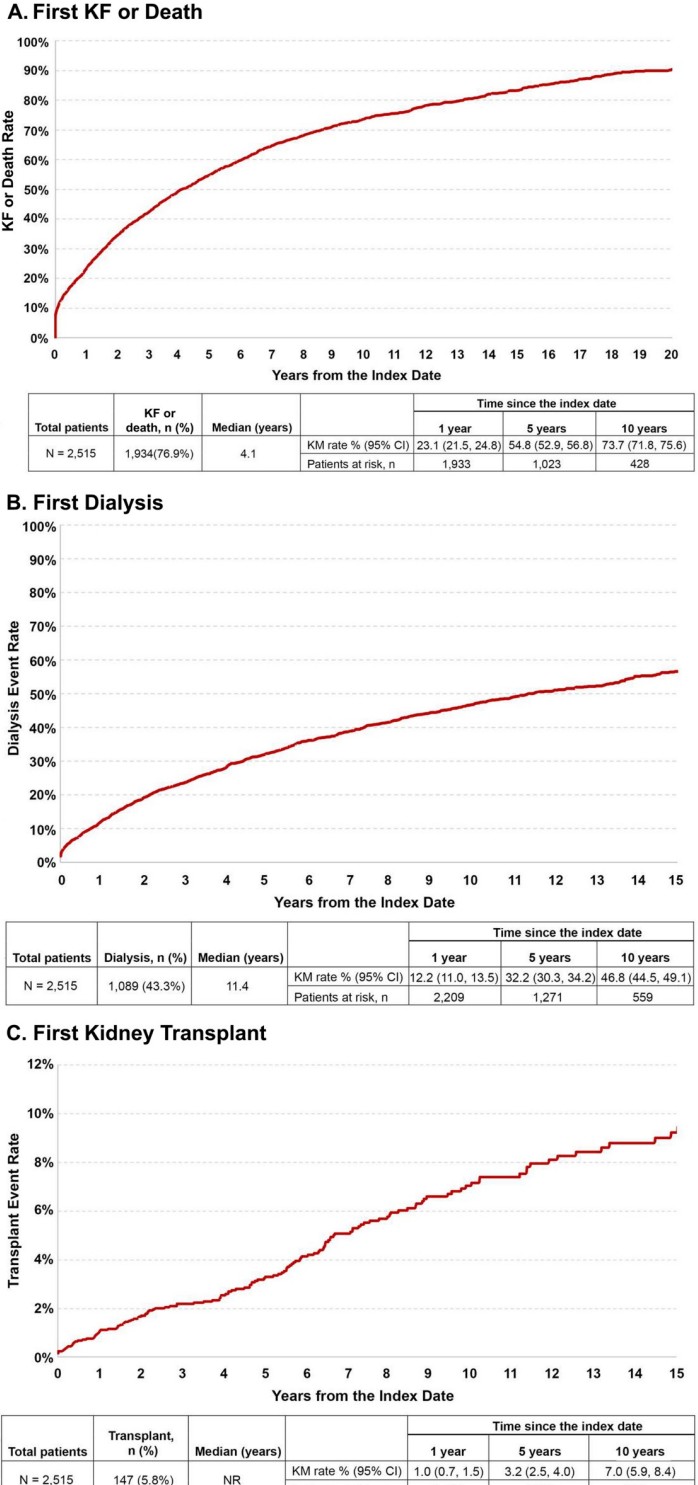

**Fig 3. Kaplan Meier curves of time to (A) first KF or death[a], (B) first dialysis[b], and (C) first renal transplant from the index date[c].** Abbreviations: CI, confidence interval; CKD, chronic kidney disease; eGFR, estimated glomerular filtration rate; FSGS, focal segmental glomerulosclerosis; KF, kidney failure; KM, Kaplan-Meier; NR, not reached. Notes: [a] Time to first KF or death was measured as the number of days between the index date (date of first diagnosis of FSGS) and the date of first KF event or death during the follow up period. Patients who did not experience the event were censored at the end of the follow-up period. An event was defined as a composite measure including KF

identified via diagnosis codes (585.6x and N18.6x), CKD stage 5 (eGFR<15 mL/min/1.73 m$^2$ or diagnosis code [585.5x and N18.5]), kidney transplant, dialysis, and death. [b] Measured as the number of days between the index date and the date of the first dialysis event during the follow up period. Patients who did not experience the event were censored at the end of the follow-up period. [c] Measured as the number of days between the index date and date of first renal transplant event during follow up period. Patients who did not experience the event were censored at the end of the follow-up period.

dying during the follow-up period. The 1-year dialysis rate was 12.2%, similar to the 12.5% reported in a study of commercially-insured patients diagnosed with FSGS in Truven Market-Scan data [8]; in the current study, just under one-half initiated dialysis across the full follow-up period. Approximately 6% of patients were observed to receive a kidney transplant during follow-up. However, this is likely an underestimate as patients who advance to KF

**Table 4. HRU and healthcare costs during the follow-up period[a].**

| | 1$^{st}$ Year Post-Index | 2$^{nd}$ Year Post-Index[b] |
|---|---|---|
| | **N = 2,515** | **N = 2,344** |
| **All-cause HRU** | | |
| IP admissions, n (%) | 1,001 (39.8) | 617 (26.3) |
| Number of visits, mean (SD) [median] | 0.9 (1.5) [0.0] | 0.5 (1.2) [0.0] |
| Hospital days[c], mean (SD) [median] | 19.1 (36.8) [8.0] | 17.7 (38.2) [7.0] |
| OP visits, n (%) | 2,510 (99.8) | 2,301 (98.2) |
| Number of visits, mean (SD) [median] | 33.6 (30.8) [25.0] | 30.9 (35.2) [21.0] |
| ER visits, n (%) | 830 (33.0) | 664 (28.3) |
| Number of visits, mean (SD) [median] | 0.7 (1.4) [0.0] | 0.6 (1.5) [0.0] |
| **All-cause healthcare costs (2021 USD)** | | |
| Total costs, mean (SD) [median] | 36,543 (88,858) [12,354] | 29,834 (58,025) [10,132] |
| Pharmacy costs | 5,402 (15,499) [1,505] | 4,440 (11,238) [1,406] |
| Total medical costs | 31,141 (80,422) [10,004] | 25,394 (52,046) [7,732] |
| IP admission costs | 13,655 (72,132) [0] | 8,535 (36,014) [0] |
| OP visit costs | 16,072 (24,463) [8,501] | 15,657 (28,456) [7,011] |
| ER visit costs | 1,415 (3,630) [0] | 1,202 (3,533) [0] |
| **Conditional costs (2021 USD)** | | |
| Among patients with at least 1 IP admission | N = 1,001 | N = 617 |
| IP admission costs, mean (SD) [median] | 34,308 (111,226) [7,615] | 32,425 (64,479) [10,203] |
| Among patients with at least 1 OP visit | N = 2,510 | N = 2,301 |
| OP visit costs, mean (SD) [median] | 16,103 (24,478) [8,542] | 15,944 (28,641) [7,227] |
| Among patients with at least 1 ER visit | N = 830 | N = 664 |
| ER visit costs, mean (SD) [median] | 3,770 (5,267) [1,967] | 3,747 (5,528) [1,812] |

**Abbreviations**: ER, emergency room; HRU, healthcare resource use; FSGS, focal segmental glomerulosclerosis; IP, inpatient; OP, outpatient; SD, standard deviation; USD, United States dollars; VA, Veterans Affairs.

**Notes**:

[a] Analyses included HRU and healthcare costs observable in the VA data, which includes only HRU that occurred within VA facilities and costs as reported in the VA database (i.e., paid by the VA). Any medical visits that occurred in other medical facilities, and costs that were covered by other insurance (e.g., Medicare, Medicaid, or private insurance) were not observable in the data.

[b] Among patients who had ≥2 years of continuous eligibility following the index date (the date of first diagnosis of FSGS).

[c] Calculated among patients with ≥1 inpatient admission during the relevant time period.

automatically become eligible for Medicare, and thus may receive a transplant outside the VA which would not be observable in the data.

VA patients with a diagnosis associated with FSGS utilized substantial healthcare resources, with 40% and 33% of patients having ≥1 inpatient admission and/or emergency room visit, respectively, during the first year after the index date. In addition, patients had on average 34 outpatient visits during the first year post-index. Healthcare costs to the VA were high during the first year of follow-up, with a mean total of $36,543 per patient. Similar to the findings of this study, Kalantar-Zadeh et al. also reported that patients with FSGS in the Optum Clinformatics Data Mart Database, which includes patients with commercial or Medicare Advantage insurance in the US, bore a substantial clinical burden [7]. In that study, 28.8% of patients experienced hospitalization and 43% visited the emergency room within a year of an index FSGS diagnosis [7]. Similarly, a study by Nazareth et al. of commercially-insured patients diagnosed with FSGS in Truven MarketScan data found that 21% and 29% of patients had inpatient admissions and emergency room visits, respectively, during the first year after the initial FSGS diagnosis [8].

This study utilized VA data, which is a rich data source that allows for the assessment of a large population with a long follow-up time. However, there are some limitations associated with this data. Patients included in the VA represent a unique population whose epidemiology, characteristics, and outcomes may not apply to the general population; in particular, the population is primarily male. Medical services provided outside of VA facilities were not observable in the data and may result in misclassification of incident and prevalent patients with associated under estimation of the incidence and prevalence of FSGS in the VA population. As most people in the VA system receive care at VA facilities, the effects of this would likely be minimal with the exception of those who progress to KF. Patients diagnosed with KF become eligible for Medicare coverage and may receive care–in particular, dialysis and renal transplant–outside of the VA system, which would not be recorded in the VA data. Additionally, due to limited capacity at VA-based dialysis centers, patients may be more likely to seek this care outside of the VA. One study found that only 10% of veterans began dialysis treatment in a VA-based dialysis center when they had advanced CKD [24]. Therefore, KF treatments such as dialysis and renal transplant, KF-related clinical outcomes, and associated healthcare costs are likely to be underestimated. The costs available in the database represent healthcare costs paid by the VA and may not be generalizable to other payers such as commercial insurance or Medicare; further, these healthcare cost results may not be comparable to those reported in other studies. In addition, indirect costs, such as work loss or loss of productivity, were not observable in the data. FSGS was identified by diagnosis codes which did not differentiate between primary and secondary FSGS. Additionally, the database lacked clinical information such as disease severity or progression. Finally, although the treatment landscape for FSGS did not substantially change during the study period, increased patient or physician awareness of the disease could have impacted the outcomes assessed in this study.

## Conclusions

Among US veterans, FSGS was found to be associated with considerable clinical and economic burdens, including progression to KF and high healthcare costs. This highlights the need for improved FSGS treatment options that delay or prevent progression to KF, as well as minimize the HRU and cost burdens to patients and healthcare systems.

## Supporting information

**S1 Fig. Sample selection.** Data Source: Veteran Affairs Health System Corporate Data Warehouse (October 1999—February 2021). Abbreviations: COVID-19, coronavirus 2019; FSGS,

focal segmental glomerulosclerosis; ICD-9/10- CM, International Classification of Diseases, Ninth/Tenth Revision, Clinical Modification.
(TIF)

**S1 Table. Medication used during the follow-up period[a], by drug class: United States, 2000–2020.** Abbreviations: ACE, angiotensin-converting enzyme; FSGS, focal segmental glomerulosclerosis; MRA, mineralocorticoid receptor antagonist; SD, standard deviation; SGLT2, sodium-glucose transport protein 2. Notes: [a] The follow-up period spanned from the index date (the date of first diagnosis date of FSGS) to the end of enrollment date, end of data availability, or date of death, whichever occurred first. [b] Among patients who had ≥2 years or ≥3 years of continuous eligibility following the index date.
(DOCX)

**S1 Checklist. STROBE statement—Checklist of items that should be included in reports of *cohort studies*.**
(PDF)

## Acknowledgments

Medical writing assistance was provided by Shelley Batts, PhD, an independent contractor of Analysis Group, Inc., and paid for by Travere Therapeutics, Inc.

## Author Contributions

**Conceptualization:** Deborah Goldschmidt, Mark E. Bensink, Zheng-Yi Zhou, Sherry Shi, Yilu Lin, Lizheng Shi.

**Data curation:** Yilu Lin, Lizheng Shi.

**Formal analysis:** Sherry Shi, Yilu Lin.

**Funding acquisition:** Mark E. Bensink.

**Investigation:** Deborah Goldschmidt, Mark E. Bensink, Zheng-Yi Zhou, Sherry Shi, Yilu Lin, Lizheng Shi.

**Methodology:** Deborah Goldschmidt, Mark E. Bensink, Zheng-Yi Zhou, Sherry Shi, Yilu Lin, Lizheng Shi.

**Project administration:** Deborah Goldschmidt, Mark E. Bensink, Zheng-Yi Zhou, Sherry Shi, Yilu Lin, Lizheng Shi.

**Resources:** Mark E. Bensink.

**Software:** Sherry Shi, Yilu Lin.

**Supervision:** Deborah Goldschmidt, Mark E. Bensink, Zheng-Yi Zhou, Lizheng Shi.

**Validation:** Sherry Shi, Yilu Lin.

**Writing – original draft:** Deborah Goldschmidt, Mark E. Bensink, Zheng-Yi Zhou, Sherry Shi, Yilu Lin, Lizheng Shi.

**Writing – review & editing:** Deborah Goldschmidt, Mark E. Bensink, Zheng-Yi Zhou, Sherry Shi, Yilu Lin, Lizheng Shi.

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
