## [Decision Letter · Decision Letter 0]

11 Sep 2024

PONE-D-24-22665Epidemiology and Burden of Focal Segmental Glomerulosclerosis among United States Veterans: An Analysis of Veteran’s Affairs DataPLOS ONE

Dear Dr. Bensink,

Thank you for submitting your manuscript to PLOS ONE. After careful consideration, we feel that it has merit but does not fully meet PLOS ONE’s publication criteria as it currently stands. Therefore, we invite you to submit a revised version of the manuscript that addresses the points raised during the review process.

We look forward to receiving your revised manuscript.

Kind regards,

Meng Li

Academic Editor

PLOS ONE

Journal Requirements:

2.Thank you for stating the following in the Competing Interests/Financial Disclosure * (delete as necessary) section:

"Funding for this study was provided by Travere Therapeutics, Inc. The sponsor was involved in the study design, data analysis, decision to publish, and preparation of the manuscript."

We note that you received funding from a commercial source: "Travere Therapeutics Inc"

3. In the online submission form, you indicated that "The data in this study are not publicly available and cannot be shared, although all the National VA Health Care Network data are available by request from the US Department of Veterans Affairs."

Reviewers' comments:

Reviewer's Responses to Questions

**Comments to the Author**

1. Is the manuscript technically sound, and do the data support the conclusions?

Reviewer #1: Yes

Reviewer #2: Yes

2. Has the statistical analysis been performed appropriately and rigorously? 

Reviewer #1: Yes

Reviewer #2: Yes

3. Have the authors made all data underlying the findings in their manuscript fully available?

Reviewer #1: Yes

Reviewer #2: No

4. Is the manuscript presented in an intelligible fashion and written in standard English?

Reviewer #1: Yes

Reviewer #2: Yes

5. Review Comments to the Author

Reviewer #1: Overall I found the manuscript to be well written. I have some minor comments/questions below. I hope that my comments help improve the clarity and depth of the content.

The patient selection period spanned over 20 years and treatment landscape may have changed significantly over the years. For example, patients indexed in more recent years could have better outcomes because of new treatments, better awareness etc. How did you address this in your analysis for OS and time to events? How does this affect HRU and costs, especially among subgroups with 2+ years of follow-up?

111: What is the rationale of including death as kidney failure? Wouldn't that overestimate the incidence of KF?

120: Is there a washout period for incident patients? How did you make sure that the first FSGS diagnosis observed during each year in VA to be the "true" first diagnosis in real life?

121: "Prevalence was estimated as the number of patients with follow-up..." Is there a specific length of follow-up required here?

278: External validity: other than having a higher proportion of black/male patients, are the VA data representative of the general commercially insured patients in age, comorbidities, coverage, and access to care?

Reviewer #2: The introduction mentions current treatment options like glucocorticoids and calcineurin inhibitors, but the use of calcineurin inhibitors or other disease-modifying medications were not included in baseline characteristics. Consider providing more details or explanation of why patients were not on calcineurin inhibitors or other disease-modifying agents.

In Table 1, report the units for eGFR (mL/min/1.73 m²), clarify how eGFR was calculated.

Include units for eGFR in the Results section, specifically under section 3.2, Patient Characteristics.

Consider putting clinical outcomes such as incidence of cardio/cerebrovascular events, nephrotic syndrome, and death in its own table.

Consider including the Kaplan-Meier cumulative incidence curves and survival curves within the main text and summarize the results in a table.

Since this is a retrospective cohort study, consider including the STROBE Checklist in the supplemental, which outlines essential items for reporting cohort studies.

Table 3: Healthcare Costs

Clarify whether the conditional costs are also adjusted to 2021 USD$. For all-cause healthcare costs and conditional costs, consider including the 25th to 75th percentiles.

Section 3.7: Healthcare Costs During Follow-Up

Clarify the meaning of the numbers included in brackets, such as pharmacy costs (mean: $5,402 [$15,499]) and medical costs (mean: $31,141 [$80,422]). Ensure consistency between the text and the table, where "$15,499" and “$80,422”are indicated as the standard deviation in parentheses in table 3, but appears as the median in brackets in the text.

Discussion

Consider mentioning the trend/ implication of the trend in the prevalence and incidence of PSGS patients. Compare prevalence reported in this paper with the global/ US prevalence of PSGS.

6. PLOS authors have the option to publish the peer review history of their article (what does this mean?). If published, this will include your full peer review and any attached files.

Reviewer #1: **Yes: **Zifan Zhou

Reviewer #2: No

---

## [Author Response · Author response to Decision Letter 0]

27 Oct 2024

Comments from Academic Editor:

Journal Requirements:

Author response: Thank you for letting us know. The files have been updated per the templates you noted, and the files have been renamed.

2.Thank you for stating the following in the Competing Interests/Financial Disclosure * (delete as necessary) section:

"Funding for this study was provided by Travere Therapeutics, Inc. The sponsor was involved in the study design, data analysis, decision to publish, and preparation of the manuscript."

We note that you received funding from a commercial source: "Travere Therapeutics Inc"

Author Response: Thank you, we have included the amended Competing Interests statement in the cover letter. 

3. In the online submission form, you indicated that "The data in this study are not publicly available and cannot be shared, although all the National VA Health Care Network data are available by request from the US Department of Veterans Affairs."

Author Response: Thank you for noting this. The raw data used in this study was provided under an agreement with the National VA Health Care Network. The data cannot be shared publicly because the Department of Veterans Affairs prevents public sharing of national VA EHR data. Data are available only to VA investigators and interested researchers can request access to the data. Therefore, we respectfully request an exemption from this requirement.

We have updated the Data Availability Statement to: “Data cannot be shared publicly because the Department of Veterans Affairs prevents public sharing of national VA EHR data. Researchers with VA appointments can request access to the data through the VA intranet at the VA Data Access Portal. For further assistance, please contact the VA Informatics and Computing Infrastructure (VINCI) at VINCI@va.gov or Yilu Lin at yilu.lin@va.gov.”

Author Response: We have reviewed the references and, to our knowledge, none of the cited publications have been retracted.

Reviewers' comments

Reviewer #1: 

Overall I found the manuscript to be well written. I have some minor comments/questions below. I hope that my comments help improve the clarity and depth of the content.

Author Response: Thank you for your time reviewing our manuscript. We appreciate your detailed suggestions which helped to strengthen the article.

The patient selection period spanned over 20 years and treatment landscape may have changed significantly over the years. For example, patients indexed in more recent years could have better outcomes because of new treatments, better awareness etc. How did you address this in your analysis for OS and time to events? How does this affect HRU and costs, especially among subgroups with 2+ years of follow-up?

Author Response: Thank you for the insightful comment. There is currently no cure for FSGS, nor any therapy specifically approved by the FDA for its treatment. Additionally, the medications that patients with FSGS receive (i.e., corticosteroids, ACE/ARBs) have been around for many decades and, unfortunately, there have been no novel therapies introduced during the study period. Although it is possible that there have been changes in awareness over time, this is difficult to quantify or adjust for. The sparse literature on patients with FSGS and intermittent updates to clinical treatment guidelines do not suggest substantial increases in awareness. However, we have added a statement in the limitations (page 24 of the tracked manuscript):

“Finally, although the treatment landscape for FSGS did not substantially change during the study period, increased patient or physician awareness of the disease could have impacted the outcomes assessed in this study. “

We agree that this will become an important question when the treatment landscape for FSGS begins to change following the entry of novel treatments. Future research could potentially examine different cohorts by year of diagnosis to compare patient outcomes before versus after a pivotal change in the treatment landscape.

111: What is the rationale of including death as kidney failure? Wouldn't that overestimate the incidence of KF?

Author Response: Thank you for the question. We apologize for the confusion on this point and have now clarified in the manuscript and figures that this refers to “kidney failure or death.” 

120: Is there a washout period for incident patients? How did you make sure that the first FSGS diagnosis observed during each year in VA to be the "true" first diagnosis in real life?

Author Response: Thank you for the question. For the incidence calculation, we used the first observed FSGS diagnosis in the data; however, we also included the requirement that a second diagnostic code be present in the data 30 to 180 days after the first observed diagnosis as a mechanism to ensure accurate identification of patients with FSGS. It is possible that patients may have had an earlier FSGS diagnosis before they entered the VA database, but this would not be captured in our analysis and is a recognized limitation of real-world data. Regarding “washout”, as our study was non-interventional, a medication “washout” is not part of the current study design.

We have now edited the following limitation on page 23:

“Medical services provided outside of VA facilities were not observable in the data and may result in misclassification of incident and prevalent patients with associated under estimation of the incidence and prevalence of FSGS in the VA population.” 

121: "Prevalence was estimated as the number of patients with follow-up..." Is there a specific length of follow-up required here?

Author Response: Thank you for the question. Patients with follow-up during the full calendar year were included in the prevalence estimate for the year; patients who died during the calendar year were also included. We have now revised the Methods section (page 8) as follows:

“Prevalence was estimated as the number of patients with follow-up during the full calendar year who had ≥2 diagnoses for FSGS 30-180 days apart, with the first diagnosis in or before the given year, divided by the number of VA-enrolled veterans in that year. Patients who met the diagnosis criteria but died during the calendar year were also included in the numerator.”

278: External validity: other than having a higher proportion of black/male patients, are the VA data representative of the general commercially insured patients in age, comorbidities, coverage, and access to care?

Author Response: Thank you for the question. The VA population reflects the specific demographic characteristics of US veterans, which is mostly male and has an older mean age compared to the general population. We noted that this patient population is unique in the limitations section (page 23):

“Patients included in the VA represent a unique population whose epidemiology, characteristics, and outcomes may not apply to the general population; in particular, the population is primarily male.”

We have also now added the below text as a new paragraph in the Discussion on page 21:

“Few studies have reported the demographic and clinical characteristics of patients with FSGS so there are limited opportunities for comparison. Among them, the findings have been variable and there are large differences across groups by geographic location, sex, and race/ethnicity [20]. For example, FSGS is 1.5-2 times more common in males vs. females and as much as 5 times more common in Black vs. White patients [2, 18, 21]. However, the main comorbidities observed among the patients in our study are generally consistent with prior literature on commercial or registry populations with FSGS. For example, the most common comorbidity in this study was hypertension (87%), similar to reports by Thomas et al. (74%; a registry of patients in the south-east US) [22], Sim et al. (75%; a commercially-insured population in Southern California) [20], and Tuttle et al. (74%; a cohort study of kidney biopsies from 43 US states) [21]. Additionally, 26% of patients in this study had diabetes, similar to Sim et al. (28%), and the mean eGFR was 43.9 ± 26.6 mL/min/1.73 m2, within the range for patients with FSGS reported by Sim et al. (40.6 ± 28.2 mL/min/1.73 m2) and a claims study using Optum data by Bensink et al. (45.8 ± 39.4 mL/min/1.73 m2) [23]. In all these prior studies, male and Black patients with FSGS were over-represented in comparison with the general US population, although to a lesser extent than in the VA data. The mean age of our patients (58 years) was older than the populations in these studies (i.e., 48 years in Bensink et al., 49 years in Thomas et al. and Tuttle et al., 51 years in Sim et al.), reflecting the generally older age of VA patients.”

Regarding healthcare access, US veterans are universally covered by the VA system as well as community healthcare providers (without charge), which differs from the healthcare access scheme for commercially-insured patients but is not unlike Medicaid- or Medicare-insured patients in the US. 

Reviewer #2: 

The introduction mentions current treatment options like glucocorticoids and calcineurin inhibitors, but the use of calcineurin inhibitors or other disease-modifying medications were not included in baseline characteristics. Consider providing more details or explanation of why patients were not on calcineurin inhibitors or other disease-modifying agents.

Author Response: Thank you for the comment. The full list of medications used by patients can be found in Table S1. For brevity, we reported the top 5 most used medications in Table 1. Calcineurin inhibitors were used by 4.8% of patients (so not in the top 5). 

In Table 1, report the units for eGFR (mL/min/1.73 m²), clarify how eGFR was calculated.

Author Response: Thank you for noting this. We have now added the units for eGFR to Table 1. The eGFR values were taken directly from the VA laboratory data, and details about the calculation method are not available. We have added a footnote about this to Table 1:

“eGFR values were summarized based on what was directly provided in the VA laboratory data; no details on the eGFR calculation methodology were available.” 

Include units for eGFR in the Results section, specifically under section 3.2, Patient Characteristics.

Author Response: Thank you, we have now added it there (page 12).

“Among the 1,505 (59.8%) with a baseline eGFR measurement, mean eGFR level was 43.9 (SD: 26.6; median: 38.0) mL/min/1.73 m².”

Consider putting clinical outcomes such as incidence of cardio/cerebrovascular events, nephrotic syndrome, and death in its own table.

Author Response: Thank you for the suggestion. We have now separated those results in Table 2. 

Consider including the Kaplan-Meier cumulative incidence curves and survival curves within the main text and summarize the results in a table.

Author Response: Thank you for the suggestion. We have now moved those results to the main figures (now Figures 2 and 3) and included a summary table (new Table 3).

Since this is a retrospective cohort study, consider including the STROBE Checklist in the supplemental, which outlines essential items for reporting cohort studies.

Author Response: Thank you, we have now included the STROBE Checklist as a supplemental item.

Table 3: Healthcare Costs

Clarify whether the conditional costs are also adjusted to 2021 USD$. For all-cause healthcare costs and conditional costs, consider including the 25th to 75th percentiles.

Author Response: All costs were adjusted to 2021 US dollars, and we have now clarified that applies to conditional costs in Table 3. We have included mean with standard deviation as well as median costs, therefore for brevity in the table we have not included the 25th and 75th percentiles. 

Section 3.7: Healthcare Costs During Follow-Up

Clarify the meaning of the numbers included in brackets, such as pharmacy costs (mean: $5,402 [$15,499]) and medical costs (mean: $31,141 [$80,422]). Ensure consistency between the text and the table, where "$15,499" and “$80,422”are indicated as the standard deviation in parentheses in table 3, but appears as the median in brackets in the text.

Author Response: Thank you for the comment. We have revised that section for clarity as you suggest (i.e., reporting standard deviation within parentheses like in the table). 

Discussion

Consider mentioning the trend/ implication of the trend in the prevalence and incidence of PSGS patients. Compare prevalence reported in this paper with the global/ US prevalence of PSGS.

Author Response: Thank you for the suggestion. The global prevalence of FSGS is not well understood and to our knowledge there are no reliable studies reporting this data. There is more evidence for the incidence of FSGS, which we discussed in the first paragraph of the Discussion (page 18). We have added additional text as follows:

“The US incidence of FSGS as estimated by Kitiyakara et al. in 2003 was about 7 per 1 million people [18], while the incidence observed in this study was 26 per 1 million US veterans in 2003.”

Regarding the trends over time, we have now added the following paragraph to the Discussion (page 20):

“FSGS incidence observed in this study varied from 41 per 1 million in 2000 to 10 per 1 million in 2020. There was less variation in the incidence between 2001-2015; thereafter, the incidence generally declined to its lowest point in 2020. The prevalence of FSGS increased over time (from 62 per 1 million in 2000 to approximately 220 per million veterans in 2020), although this was primarily due to new patients being added each year and prior patients remaining in the data. It is possible that the stable/dec

---

## [Editor Report · Decision Letter 1]

25 Nov 2024

Epidemiology and Burden of Focal Segmental Glomerulosclerosis among United States Veterans: An Analysis of Veteran’s Affairs Data

PONE-D-24-22665R1

Dear Dr. Bensink,

We’re pleased to inform you that your manuscript has been judged scientifically suitable for publication and will be formally accepted for publication once it meets all outstanding technical requirements.

Kind regards,

Meng Li

Academic Editor

PLOS ONE
---

## [Editor Report · Acceptance letter]

4 Dec 2024

PONE-D-24-22665R1 

PLOS ONE

Dear Dr. Bensink, 

I'm pleased to inform you that your manuscript has been deemed suitable for publication in PLOS ONE. Congratulations! Your manuscript is now being handed over to our production team.

Kind regards, 

on behalf of

Dr. Meng Li 

Academic Editor

PLOS ONE